# Heterogeneous Knowledge for Augmented Modular Reinforcement Learning

**Lorenz Wolf**                                                                 *lorenz.wolf.22@ucl.ac.uk*
*Department of Computer Science & Centre for Artificial Intelligence*
*University College London*

**Mirco Musolesi**                                                              *m.musolesi@ucl.ac.uk*
*Department of Computer Science & Centre for Artificial Intelligence*
*University College London*
*Department of Computer Science and Engineering*
*University of Bologna*

**Reviewed on OpenReview:** *https://openreview.net/forum?id=eme87YbiND*

## Abstract

Existing modular Reinforcement Learning (RL) architectures are generally based on reusable components, also allowing for "plug-and-play" integration. However, these modules are homogeneous in nature. In fact, they essentially provide policies obtained via RL through the maximization of individual reward functions. Consequently, such solutions still lack the ability to integrate and process multiple types of information (i.e., heterogeneous knowledge representations), such as rules, sub-goals, and skills from various sources. In this paper, we discuss several practical examples of heterogeneous knowledge and propose Augmented Modular Reinforcement Learning (AMRL) to address these limitations. Our framework uses a selector to combine heterogeneous modules and seamlessly incorporate different types of knowledge representations and processing mechanisms. Our results demonstrate the performance and efficiency improvements, also in terms of generalization, which can be achieved by augmenting traditional modular RL with heterogeneous knowledge sources and processing mechanisms. Finally, we examine the safety, robustness, and interpretability issues stemming from the introduction of knowledge heterogeneity.

## 1 Introduction

Reinforcement learning (RL) has emerged as a powerful paradigm for sequential decision-making in complex and dynamic environments (Sutton & Barto, 2018; Kiran et al., 2022). Its successes extend across a wide range of domains, including robotics (Kober et al., 2013), dynamic resource allocation (Waschneck et al., 2018), and game playing (Mnih et al., 2015; Silver et al., 2016; Vinyals et al., 2019). Despite these advances, the practical deployment of RL in real-world systems remains constrained by two fundamental challenges: low sample efficiency and safety concerns arising from exploratory behaviors that may lead to high-risk actions in critical settings (García & Fernández, 2015).

One approach to mitigating sample inefficiency is modular RL, in which tasks are decomposed into sub-problems and solved by specialized skill modules. These modules can be reused across contexts, accelerating learning and improving flexibility (Jacobs et al., 1991; Russell & Zimdars, 2003; Sprague & Ballard, 2003; Simpkins & Isbell, 2019; Devin et al., 2017). Prior modular and hierarchical RL (HRL) frameworks typically assume *homogeneous knowledge* among modules. That is, most sub-policies are trained via RL under comparable reward structures. This reliance on homogeneity presents both theoretical and practical barriers to incorporating diverse knowledge sources. For example, a rule-based controller, a retrieval mechanism, and an RL-trained policy may differ not only in their representations and processing pipelines but also in their

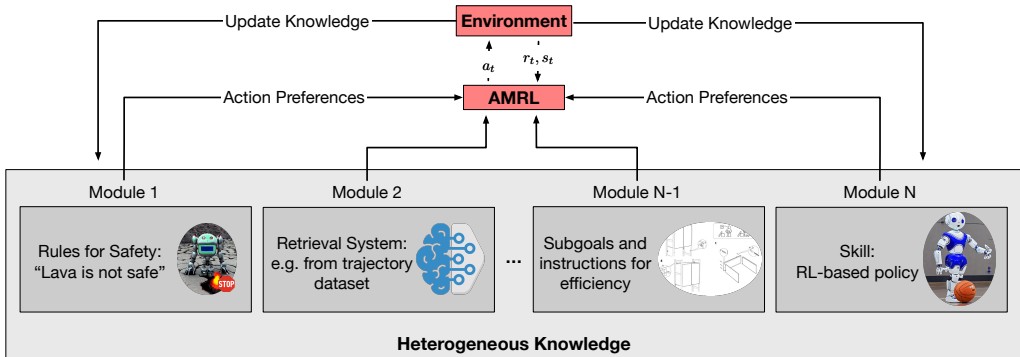

Figure 1: Examples of heterogeneous knowledge and the AMRL architecture. AMRL is able to access several sources of heterogeneous knowledge via modules. The modules can then be updated based on the environment feedback.

action-value semantics, making direct integration challenging without architectural modifications. Existing frameworks such as KoGuN (Zhang et al., 2020) and KIAN (Chiu et al., 2023) illustrate this limitation: while modular in structure, they are not explicitly designed to accommodate modules with fundamentally different internal representations, leading to performance degradation when heterogeneity is introduced.

However, in practice, diverse knowledge sources are often readily available and highly complementary. Rules and logical constraints can encode safety-critical requirements; pre-trained skills can provide strong priors; retrieval mechanisms can recall past scenarios; and RL modules can adapt online to optimize performance. Leveraging this *heterogeneous knowledge*, i.e., information derived from distinct paradigms, each with its own representations and processing mechanisms, has the potential to enhance sample efficiency, safety, and robustness simultaneously. While previous work has explored integrating single alternative sources, such as text manuals (Zhong et al., 2020), demonstrations (Abbeel et al., 2010), or knowledge graphs (Murugesan et al., 2021), these efforts remain constrained by their focus on one type of knowledge at a time. There is currently no general framework that flexibly and systematically integrates diverse sources of heterogeneous knowledge.

To address this gap, we propose Augmented Modular Reinforcement Learning (AMRL)[1] (Figure 1), a unifying framework that is explicitly designed to support the seamless integration of heterogeneous knowledge modules. In AMRL, all modules consume the same environment state and act within the same action space, but critically, they process the state differently: rule-based modules extract abstract features (e.g., "is there lava ahead?"), retrieval modules rely on stored experiences, and skill modules employ task-specific learned representations. This ensures that heterogeneity is not superficial but reflects genuinely distinct internal interpretations of the environment. AMRL introduces a general selector that arbitrates among modules through two mechanisms: *hard selection*, corresponding to command arbitration (Simpkins & Isbell, 2019), and *soft selection*, akin to command fusion (Russell & Zimdars, 2003) and also Mixture-of-Experts (Shazeer et al., 2017). While command arbitration and command fusion are not novel frameworks in isolation, we deploy these selection mechanisms to incorporate heterogeneous knowledge and show that their behaviors differ significantly in the presence of heterogeneous modules: soft selection, in particular, improves robustness and mitigates the negative impact of noisy or uninformative modules.

As a motivating example, consider an autonomous traffic control system at a busy intersection. An effective system would need to: (1) follow traffic laws expressed as rules, (2) reuse pre-trained policies from similar intersections, (3) optimize flow dynamically through RL, and (4) retrieve past cases to handle emergencies. Such a system inherently requires heterogeneous modules with distinct representations and reasoning mechanisms. AMRL enables precisely this type of integration, providing a principled architecture that abstracts away representational differences while ensuring safe and efficient operation.

---

[1]The full implementation and code used for the experiments are publicly available: `https://github.com/lorenzflow/amrl`.

In summary, we make the following main contributions:

- *Unified formalization of heterogeneous modules for decision-making.* We introduce a framework for representing and integrating diverse knowledge sources—rules, logic, skills, and retrievals—within a modular RL system.

- *Augmentation of modular RL.* We extend classical modular RL by enabling plug-and-play integration of heterogeneous modules through a selector mechanism that is agnostic to internal representations, thereby improving sample efficiency, safety, and robustness.

- *Comprehensive evaluation of heterogeneous knowledge integration.* We empirically demonstrate that AMRL not only improves sample efficiency but also enhances robustness, safety, generalizability, and interpretability. This evaluation offers new insights into the benefits and trade-offs of modular approaches when enriched with external knowledge.

In this regard, AMRL represents a substantive augmentation of modular RL, offering an empirically-validated framework that explicitly and systematically integrates heterogeneous knowledge.

## 2 Related Work

**Knowledge augmented RL.** Grounding RL agents with external knowledge is an ongoing line of research in the field of RL. For example, progress has been made on incorporating knowledge provided to the agent via text for example as manual for a game (Zhong et al., 2020) or knowledge graphs connecting concepts and their characteristics (Murugesan et al., 2021). One approach to incorporate knowledge sources is the formalism of a contextual MDP (CMDP) (Kirk et al., 2023; Perez et al., 2020; Ghosh et al., 2021; Hallak et al., 2015). Other approaches equip agents with retrieval mechanisms to access a memory of past trajectories to inform current decisions (Goyal et al., 2022; Humphreys et al., 2022).
While these methods demonstrate the value of augmenting RL with external knowledge, they remain limited in scope, integrating only a single type of knowledge (e.g., trajectories, rules, or symbolic abstractions). They typically assume that knowledge can be processed within a uniform framework. Hybrid approaches, such as the hierarchical planning method by Lorang et al. (2024) and symbolic recovery in Goel et al. (2022), partially address broader knowledge integration but are domain-specific. Modular frameworks like KoGuN (Zhang et al., 2020) and KIAN (Chiu et al., 2022; 2023) support multiple modules, but they still assume relative homogeneity, either in representation or in how modules are trained and combined. In contrast, AMRL explicitly formalizes the integration of heterogeneous knowledge sources, including rules, retrieval modules, and pre-trained skills, through a unified interface that is agnostic to differences in representation and processing, enabling flexible composition.

**Hierarchical and Modular RL.** Hierarchical RL (HRL) decomposes a problem into several smaller subtasks organized in a hierarchy (Digney, 1998). The higher-level parent-tasks can call lower-level child-tasks as if they were primitive actions. This enables higher-level tasks to focus on more abstract and longer-term learning while lower-level tasks are responsible for primitive actions and fine-grained control (Hengst, 2010). Each task is learned by a policy and at decision time; the higher-level policy triggers one in the level below. The goal is for the agent to perform task abstraction by focusing on more high-level tasks, so that it will suffer less from the curse of dimensionality (Barto & Mahadevan, 2003). This is a very active area of research with some of the earlier works dating back to the 1990s (Dayan & Hinton, 1992; Vezhnevets et al., 2017; Parr & Russell, 1997). HRL is closely related to modular RL, which aims to decompose the learning process into complementary policies/modules that can be combined by a controller (Sutton et al., 2011; Goyal et al., 2020; Jacobs et al., 1991). Previous works in Modular RL have considered command arbitration, selecting one module, which takes over the agent for the next action, and command fusion, combining the preferences of the different modules (Gupta et al., 2021; Russell & Zimdars, 2003; Mendez et al., 2022; Andreas et al., 2017). We leverage the same modularity and flexibility for heterogeneous knowledge.

**Command arbitration and fusion.** The selector used by AMRL is closely linked to the areas of command arbitration and command fusion. In the command arbitration paradigm an arbitrator decides which module takes control over the agent in a given state; the selected module then executes an action. Earlier works include GM-Sarsa (Sprague & Ballard, 2003) and Q-decomposition (Russell & Zimdars, 2003). More recent methods Arbi-Q (Simpkins & Isbell, 2019) and GRACIAS (Gupta et al., 2021) assume that the arbitrator receives a global reward signal to be maximized, enabling it to handle different scales of rewards for each of the modules. Furthermore, the global reward signal means that these methods can be used with off-policy as well as on-policy learning (Simpkins & Isbell, 2019). The underlying mechanism in command fusion is also akin to the gating function in Mixture-of-Experts (MoE) architectures (Shazeer et al., 2017; Chen et al., 2022), which has been widely used in Deep Learning and also in Large Language Models.

These prior works focus on the arbitration and fusion of homogeneous RL policies. In contrast, AMRL investigates these mechanisms in the context of heterogeneous modules, where differences in representation, processing, and reliability introduce new challenges. Our experiments show that soft selection (fusion) can learn to mitigate conflicts and noise among diverse modules, improving robustness, while hard selection (arbitration) may amplify instability when modules provide conflicting or uninformative guidance. This highlights that classical selection strategies must be re-evaluated when extended to heterogeneous integration.

## 3 Preliminaries

Let $\mathcal{M} = (\mathcal{S}, \mathcal{A}, \mathcal{P}, R, \rho_0, \gamma, T)$ represent a discrete-time finite- horizon discounted Markov decision process (MDP). $\mathcal{M}$ consists of the following elements: the state space $\mathcal{S}$, the action space $\mathcal{A}$, a transition probability distribution $\mathcal{P} : \mathcal{S} \times \mathcal{A} \times \mathcal{S} \to \mathbb{R}_+$, a reward function $R : \mathcal{S} \times \mathcal{A} \to \mathbb{R}$, an initial state distribution $\rho_0 : \mathcal{S} \to \mathbb{R}_+$, a discount factor $\gamma \in [0, 1]$, and the horizon $T$. Generally, the policy $\pi^\theta : \mathcal{S} \times \mathcal{A} \to \mathbb{R}_+$ is parameterized by $\theta$, which we optimize in order to maximize the expected discounted return under the policy. We denote the probability of taking action $a$ in state $s_t$ by $\pi^\theta(a|s_t)$ and abuse notation by denoting the distribution over all actions with $\pi^\theta(s_t)$. Even if the architecture is conceptually independent from the underlying RL algorithm, the design and implementation presented in this paper relies on Proximal Policy Optimization (PPO) (Schulman et al., 2017) for discrete action spaces and Soft Actor-Critic (SAC) (Haarnoja et al., 2018) in the continuous case.

Moveover, in many real-world deployments, the underlying state is partially observable. Formally, these scenarios can be modeled as a partially-observable Markov Decision Processes (POMDPs), which is represented by the tuple $(\mathcal{S}, \mathcal{A}, \mathcal{P}, \Omega, \mathcal{O}, R, \rho_0, \gamma, T)$, where in addition to the already known quantities we have introduced the observation space $\Omega$ and the observation probability distribution $\mathcal{O}(\cdot)$. Instead of $s \in \mathcal{S}$, the agent now observes $o \in \Omega$, generated from the state $s$ via $o \sim \mathcal{O}(s)$.

## 4 Heterogeneous Knowledge Sources, Representation, and Processing

Heterogeneity can arise for various reasons. As a consequence, AMRL agents not only rely on *heterogeneous knowledge* sources but also on heterogeneous representation and processing mechanisms (Table 1). Knowledge from the same source can be represented in various ways and, naturally, rules must be processed differently to trajectories. Importantly, this goes beyond prior modular or hierarchical RL frameworks, which typically assume homogeneous modules consisting of RL-trained sub-policies with comparable reward scales, and therefore do not consider modules with fundamentally different representational formats or reasoning paradigms. We will adopt a bottom-up approach in the description of AMRL: we first discuss the modules at the basis of our approach; they essentially act as containers for heterogeneous knowledge sources and processing. In particular, we focus on the four types of modules outlined in the motivating example, specifically, rules, skills, retrieval, and dynamic RL. We then discuss how these modules are incorporated in the overall architecture via a selection mechanism based on hierarchical modular RL.

### 4.1 Knowledge Representation and Processing using Modules

An AMRL agent is instantiated with several modules $M_i \in \mathbf{M}$, which each act as a container for knowledge representation $K_i$ and a corresponding processing mechanism. Module $M_i$ generates action preferences,

Table 1: Knowledge heterogeneity can stem from various factors, namely the source of the knowledge, its representation, and the processing methods applied.

| Source | Representation | Processing |
|--------|----------------|------------|
| Human Expert | Rules | Logic |
| Environment | RL-based policy | Skill Execution |
| Previous Deployment | Trajectory Database | Retrieval |

represented as probabilities over possible actions, based on its knowledge source. This is expressed mathematically as $\pi_{M_i}(a|s_t) = M_i(a, s_t|K_i)$. In contrast to previous works in modular RL, the AMRL modules are heterogeneous. They are not limited to RL policies and can be diverse in terms of how they form preferences over actions. Modules and their knowledge sources can be either static or dynamic. If the knowledge source, and consequently the mapping of a module $M_i$, needs to be updated during training, modular feedback must be observed. This ensures that the effects of actions proposed by $M_i$ can be learned effectively. Modular rewards $\{r_{i,t}\}_{i=1}^N$, where $r_{i,t}$ is the reward for module $M_i$ at time $t$, can be specified to favor modular learning. In case $M_i$ is static no modular updates are performed.

We now focus on the knowledge $K_i$ contained in modules $M_i$. Knowledge sources can be represented in many different forms, including rules, manuals, trajectories, etc. and can be assessed according to the following characteristics: *actionability*, *interpretability*, and *informativeness*. A knowledge representation is *actionable* if it can be used directly to inform the action selection process. Most actionable choices are procedural or declarative knowledge sources, for example in form of a world model (Hafner et al., 2025) or a database of past trajectories (Goyal et al., 2022; Humphreys et al., 2022). While these might be the most effective representations for RL agents, they are less interpretable and importantly can be harder to obtain in the first place due to deployment restrictions. Other more abstract knowledge representations such as rules and manuals are more *interpretable*, easier to validate and in many cases more widely available, which highlights the importance of flexibly incorporating and acquiring heterogeneous knowledge. The *informativeness* captures how relevant and useful the knowledge is for solving the given task. In our evaluation we compare the effect of these properties on the agent's performance and sample efficiency.

### 4.2 Types of Modules

In this subsection, we provide practical examples of heterogeneous modules. It is important to note that the proposed architecture is flexible and supports multiple types of modules, each based on different knowledge representations and processing mechanisms. In the following, we select modules that are representative of broader classes of solutions. We first describe each module type for discrete action spaces and then outline how it can be extended to continuous action spaces. We are also aware of ongoing research on each of these (Liu et al., 2023; Likmeta et al., 2020; Hasanbeig et al., 2023; Goyal et al., 2022; Humphreys et al., 2022), but note that our focus is on combining heterogeneous decision-making mechanisms. A more detailed examination of each possible mechanism is beyond the scope of this paper.

#### 4.2.1 Logic-based Rules

This module relies on a set of rules $\Lambda = \{\lambda_i(c_i)\}_{i=1}^{N_\lambda}$, where each rule $\lambda_i(c_i)$ is formalized as statement of the form:

$$\text{If } s_t \text{ satisfies condition } c_i \text{ then } \pi_M(a|s_t) = p_{\lambda_i(c_i)}(a),$$

where $p_{\lambda_i(c_i)}(a)$ is the probability of taking action $a$ under rule $\lambda_i$ with condition $c_i$ satisfied. For example, let us consider one of the typical benchmark games where an agent has to move in a space where a lava flow is present: if there is lava in front of the agent then the probability of moving forward is set to 0. Conversely, let us suppose to have a different game environment in which, for instance, an agent has to collect objects, such as keys: if there is a key in front of the agent then the probability of picking up the key is set to 1. Rules can be organized hierarchically, with one rule potentially relying on other rules. Furthermore, rules can contradict or coincide in which case the conflict needs to be resolved. As this is not the main focus of

this work, we simply resolve such conflicts by averaging the probabilities over the set of rules applicable in state $s_t$ denoted by $\Lambda_t$, such that:

$$\pi_M(a|s_t, \Lambda) = \frac{1}{|\Lambda_t|} \sum_{\lambda \in \Lambda_t} p_{\lambda(c)}(a). \tag{1}$$

### 4.2.2 Trajectory Database with Retrieval

This module relies on a database containing trajectories and retrieves relevant information via nearest-neighbor search. The knowledge source in this case is given by the set $\mathcal{D} = \{\tau_i\}_{i=1}^{N_D}$, where each $\tau_i = ((s_{i,1}, a_{i,1}, r_{i,1}), \ldots, (s_{i,T_i}, a_{i,T_i}, r_{i,T_i}))$ is a trajectory of length $T_i$ containing state-action-reward tuples. The retrieval mechanism relies on an embedding network mapping from the state space to an embedding space. Given the current state the query is formed via $q_t = embed(s_t)$. The $k$ approximate nearest-neighbors in the embedding space are retrieved from $\mathcal{D}$ with the Facebook AI similarity search library (FAISS) (Johnson et al., 2021) based on the $L_2$-norm, which yields the k tuples $(s_{n_1}, a_{n_1}, r_{n_1}), \ldots, (s_{n_k}, a_{n_k}, r_{n_k})$ with corresponding distances $d_{n_1}, \ldots, d_{n_k}$ to the query. The retrieved information is utilized to form an action by weighting according to rewards and distance, which yields action preferences:

$$\pi_M(a|s_t, \mathcal{D}) = \frac{e^{p_D(a)}}{\sum_{a \in \mathcal{A}} e^{p_D(a)}}, \text{ with } p_D(a) = \sum_{j=1}^{k} \frac{r_{n_j}}{d_{n_j}} I(a = a_{n_j}). \tag{2}$$

We note that other more sophisticated methods for retrieval such as those proposed by Goyal et al. (2022) and Humphreys et al. (2022) can be substituted instead.

### 4.2.3 RL-based Policies (Skills)

The third class of modules we consider is that based on skills, i.e., on policies $\pi_{skill}$ trained via RL with an unknown reward function. The policies are trained on the same state space $\mathcal{S}$ with action space $\mathcal{A}_{skill} \subset \mathcal{A}$. These modules simply contain the policy and return corresponding action probabilities given by $\pi_M(a|s_t) = \pi_{skill}(a|s_t)$ if $a \in \mathcal{A}_k$ and 0, otherwise.

### 4.3 Extending Modules to Continuous Action Spaces

For continuous state and action spaces, the conditions may be expressed as inequalities or ranges, and instead of discrete probabilities, the rule produces a parametric distribution over actions, e.g., $\pi_M(a|s_t, \Lambda) = \mathcal{N}(\mu_\Lambda(s_t), \sigma_\Lambda^2(s_t))$ where $\mu_\Lambda(s_t)$ and $\sigma_\Lambda(s_t)$ can be constants or functions determined by the rule. The retrieval module can be adapted for continuous action spaces by replacing the indicator function with a kernel density estimate over the retrieved actions, yielding a smooth probability distribution. Additionally one needs to replace the softmax over discrete actions with normalization of the kernel density. Skill modules in the continuous action space return parameters of a distribution, e.g., Gaussian $\mathcal{N}(\mu_{\text{skill}}(s_t), \sigma_{\text{skill}}^2(s_t))$, instead of discrete probabilities.

## 5 Combining Heterogeneous Knowledge

To incorporate various types of knowledge sources and processing mechanisms as described in Section 4, we employ a controller / arbitrator, which we refer to as the *selector*. The goal of this component is to select/combine one or more modules. In particular, it determines which modules to trigger in a given state, thereby enabling the performance of tasks with varying levels of complexity.

We consider two variants of selection mechanisms, which correspond to command arbitration (Sprague & Ballard, 2003) and command fusion (Russell & Zimdars, 2003). The first, which we refer to as *hard selection*, restricts the weights to a one-hot vector so that exactly one module is selected and executed at each time step. The second, which we refer to as *soft selection*, relies on an attention mechanism to form a weighted average combining preferences from several modules, resulting in command fusion. Thus, by weighting the

modules, this variant of the selector is based in a sense on a mixture model. We denote the selector's policy mapping from the state space to the selector's action space by $\pi_{selector}^{\phi} : \mathcal{S} \to \mathcal{A}_{selector}$, parameterized by $\phi$. We have $\mathcal{A}_{selector} = \{M_1, \ldots, M_N\}$, such that $\pi_{selector}^{\phi}(M_i|s_t)$ is the probability of choosing module $M_i$ in state $s_t$ and expresses the selector's preference for module $M_i$. In the following, we will provide a formalization of the two selection mechanisms and discuss how to bridge the gap between the current task and what is possible by flexibly combining the modules.

## 5.1 Hard Selection

At each time step $t$, the selector observes the current state $s_t$ and selects exactly one module $M$ to execute. We then sample an action from the chosen module $a_t \sim \pi_M(s_t)$. In particular, one module is sampled according to the selector's policy $M \sim \pi_{selector}^{\phi}(s_t)$. To perform hard selection in a differentiable manner, we generate a sample vector $\mathbf{y}$ with dimension $|\mathcal{A}_{selector}| = N$ by setting:

$$y_i = \frac{\exp\left(\left(\log\left(\pi_{selector}^{\phi}(M_i|s)\right) + g_i\right)/\tau\right)}{\sum_{j=1}^{N}\exp\left(\left(\log\left(\pi_{selector}^{\phi}(M_j|s)\right) + g_j\right)/\tau\right)} \tag{3}$$

for $i = 1, \ldots, N$, where $g_1 \ldots g_N$ are i.i.d samples drawn from $Gumbel(0, 1)$, which can be sampled using an inverse transform (Jang et al., 2017). The obtained sample $\mathbf{y}$ is subsequently discretized into a one-hot vector. This ensures the selection of exactly one module at a time. A soft approximation is used for the computation of the gradients. This yields as the AMRL agent's policy $\pi(a|s_t) = \pi_M(a|s_t)$ with $M \sim \pi_{selector}^{\phi}(s_t)$. In the experiments we set the temperature $\tau = 1$. We provide an ablation analysis in Appendix F.3.

## 5.2 Soft Selection

In contrast to hard selection, soft selection combines the modules' policies with a weighted average. The action preferences $\pi_{M_i}(a|s_t)$ returned by the modules are weighted by the module preferences $\pi_{selector}^{\phi}(M_i|s_t)$ returned by the selector. In particular, we have that $\pi(a|s_t) = \sum_{i=1}^{N} \pi_{selector}^{\phi}(M_i|s_t)\pi_{M_i}(a|s_t)$. Note that $\pi_{selector}^{\phi}(s_t)$ is normalized to sum to 1.

## 5.3 Implementing the Selectors

In many applications, flexibly combining the knowledge available to the agent via the modules may not be sufficient to achieve high performance on a new task. Similarly to the inner actor in the KIAN architecture, we bridge this gap by adding a dynamic RL module. This module contains a learnable RL policy $\pi_{M_{dyn}}^{\theta} : \mathcal{S} \times \mathcal{A} \to \mathbb{R}_{+}$ trained to maximize the global reward function also observed by the selector. In practice, the selector is implemented as a neural network parameterized by parameters $\phi$, which, given the current state as input, outputs probabilities over the modules. By including the dynamic RL module, we have as AMRL policy with soft selection:

$$\pi^{\phi,\theta}(a|s) = \pi_{selector}^{\phi}(M_{dyn}|s)\pi_{M_{dyn}}^{\theta}(a|s) + \sum_{M \in \mathbf{M} \setminus M_{dyn}} \pi_{selector}^{\phi}(M|s)\pi_M(a|s), \tag{4}$$

which allows us to easily compute policy gradients w.r.t. $\phi$ and $\theta$, justifying the use of PPO to train the selector (see Appendix A for details). Note again that the selector itself is agnostic with respect to the actual knowledge representation. The entire decision-making process is presented in Algorithm 1.

## 5.4 Extending AMRL to Continuous Action Spaces

We now discuss the extension of the AMRL framework to continuous action spaces. The hard selection mechanism can easily be applied to continuous-action modules by selecting a single module's Gaussian policy at each time step. The action is then sampled from the selected distribution, rather than a discrete action. Gradients can be computed using the Gumbel-Softmax approximation over the module selection weights.

---

**Algorithm 1** Decision-making with an AMRL agent in discrete action spaces.

---

**Require:** Modules and knowledge $\mathbf{M} = \{M_i, K_i\}_{i=1}^N \cup M_{dyn}$
  $s_0 \leftarrow$ Initial state
  **for** each timestep $t = 0, ..., T$ **do**
    Get module preferences $\pi_{M_i}(s_t) \leftarrow M_i(s_t | K_i)$
    **if** Hard selection **then**
      Sample selected module $M \sim \pi_{selector}^\phi(s_t)$
      Get AMRL policy $\pi(a|s_t) = \pi_M(a|s_t)$
    **else if** Soft selection **then**
      $\pi(a|s_t) = \sum_{M \in \mathbf{M}} \pi_{selector}^\phi(M|s_t)\pi_M(a|s_t)$
    **end if**
    Sample action from AMRL policy $a_t \sim \pi(s_t)$
    Observe $s_{t+1}$ and reward $r_t$
    Update selector parameters $\phi$ and dynamic module parameters $\theta$ with PPO
  **end for**

---

Soft selection in continuous-action domains can be performed by forming the true mixture of Gaussian distributions for the modules. However, this would require computing the full weighted sum of densities, which makes sampling and entropy computation intractable and can lead to high-variance gradients. Instead, we approximate the fused policy as a single Gaussian whose mean and variance are the weighted sums of the individual modules' means and variances:

$$\mu_{\text{fused}}(s_t) = \sum_i w_i \mu_i(s_t), \quad \sigma_{\text{fused}}^2(s_t) = \sum_i w_i \sigma_i^2(s_t), \tag{5}$$

where $w_i$ are the selector weights and $\mathcal{N}(\mu_i(s_t), \sigma_i^2(s_t))$ are the individual module distributions. This approximation enables efficient sampling and stable gradient computation using the reparameterization trick, while retaining contributions from all modules. Forming a true mixture of Gaussians would require computing the logarithm of the sum of exponentials (*log-sum-exp*) over all components, which can increase gradient variance and reduce learning stability, particularly when the number of heterogeneous, partially deterministic knowledge modules is large. We adopt this single-Gaussian approximation to ensure robust and tractable learning with heterogeneous knowledge sources in continuous-action domains.

### 5.5 Module Prioritization

With heterogeneity come inherent differences between the modules. In particular modules such as the rule module may be considered safe and trustworthy, carefully designed and well understood. Consequently, while the rules may not be applicable in all cases, if applicable, they can be trusted and should be followed by the agent. This can be achieved with module prioritization. In the case of the rule module, whenever one or more of its rules are triggered, the rules act as a constraint, i.e., other modules' action preferences can be taken into account, but only as long as the agent does not violate any of the triggered rules. This can also be seen as masking of actions. In other words, for a given state $s$, the actions, which are not to be taken, are extracted from, for instance, the rule module in form of a mask defined by $\Lambda_a = 0$ if $a$ is an unsafe action and 1 otherwise. Then the final policy is given by $\Lambda_a \pi^{\phi,\theta}(a|s)$, where $\pi^{\phi,\theta}(a|s)$ is defined in Equation 4. In safety critical states, the applicable rules mask unsafe actions, which are not to be taken. This reduces the percentage of unsafe actions to 0 given that the rules cover all cases. Similarly to the soft and hard selection mechanisms, a soft prioritization mechanism based on specified weights can also be implemented.

## 6 Evaluation

In the evaluation of AMRL (and solutions for the integration of heterogeneous knowledge in RL more generally) we focus on the overall performance and sample efficiency, the dependence on knowledge informativeness and robustness against random modules, the safety gains during training and at test time, and the inter-

pretability of the selection mechanism. Unless specified otherwise explicitly, we use the acronym AMRL to denote AMRL with the soft selection mechanism.

## 6.1 Experimental Settings and Baselines

### 6.1.1 Overview

We now evaluate AMRL considering a range of modules across different benchmark environments. In particular, we train the agents on both discrete and continuous action spaces.

**Discrete Action Benchmarks (Minigrid).** We use several environments from the Minigrid suite (Chevalier-Boisvert et al., 2018), each presenting distinct challenges:

- *DoorKey*: The agent must locate and pick up a key, unlock a door, and reach a goal square. Evaluated on 6x6 and 8x8 grids.

- *LavaCrossing*: A safety-critical environment where the agent navigates through streams of lava to reach a goal. Stepping on lava terminates the episode. Evaluated on 9x9 grids with 1 or 2 lava streams.

- *Empty*: A simple navigation task where the agent moves to a goal square without obstacles. Evaluated on a 16x16 grid.

In all Minigrid environments, the agent has 7 discrete actions and observes a 7x7 grid centered on its position. Additional details on action space and visual representations are provided in Appendix B.

**Continuous Action Benchmark (OpenAI-Robotics).** For evaluation in continuous action spaces, we use the Fetch environments (Plappert et al., 2018), a set of manipulation tasks performed with a 7-DoF robot arm from the OpenAI Robotics Gym (de Lazcano et al., 2024). Tasks include reaching, pushing, sliding, and pick-and-place, which require precise control over continuous joint velocities and gripper actions.

**Modules Evaluated Across Benchmarks.** We consider instances of the module types described in Section 4.2. For Minigrid, the *rules module* (Section 4.2.1) contains a set of 6 rules relevant to the environments enforcing constraints such as "do not step onto lava" (Figure 2). Violations result in probability zero being assigned to unsafe actions. The *retrieval module* (Section 4.2.2) relies on expert trajectories. Lastly, the *RL-based skill module* (Section 4.2.3) leverages pre-trained policies specific to the skill environment. For the Fetch environment on the other hand, the first module moves the gripper closer to the object, while the second module contains a rule-based policy to approach the target position.
While all modules receive the same raw environment observation, the way they interpret and process this input differs substantially. Each module applies its own internal transformations and abstractions: the rule-based module distills the state into interpretable features like "is there lava ahead?", whereas the skill and retrieval modules employ learned state representations optimized for their specific tasks. This heterogeneity in internal processing allows the system to combine multiple perspectives on the environment, leveraging both structured, human-designed rules and flexible, learned behaviors, even though the external observation interface remains unified. All experiments are conducted with 10 random seeds[2], and we report mean and sample standard deviation.

### 6.1.2 Baselines

As baselines, we implement **KIAN** (Chiu et al., 2023), **KoGuN** (Zhang et al., 2020), and standard reinforcement learning algorithms: **PPO** (Schulman et al., 2017) for discrete action spaces, and **SAC** (Haarnoja et al., 2018) for continuous action spaces. KIAN and KoGuN are two cutting-edge methods for knowledge-augmented RL and are adapted to our scenario, which involves heterogeneous knowledge, to make them as competitive as possible. The implementations of all agents with discrete action spaces rely on the

---

[2]Experiments on the robotics environment are based only 4 seeds at the moment, the remaining seeds are being collected.

> ▶ If there is a key inside the state, then pick it up if possible or else move towards it.
>
> ▶ If there is a ball inside the state, then pick it up if possible or else move towards it.
>
> ▶ If there is a door inside the state, then open it if possible or else move towards it.
>
> ▶ If there is a goal inside the state, then move towards it.
>
> ▶ If there is Lava in front of the agent, do not move forward.
>
> ▶ If there is a wall in front of the agent, do not move forward.

Figure 2: Set of rules used in the MiniGrid evaluation.

`rl-starter-files` repository and `torch_ac`[3]. For continuous action spaces, SAC is implemented following standard settings from `rl-baselines3-zoo` (Raffin, 2020) and `stable-baselines3` (Raffin et al., 2021). Default hyperparameter settings are used for both PPO and SAC. Additional implementation details are provided in Appendix C.

**KIAN** learns a key for each piece of knowledge (originally rules) via an embedding layer. The actor consists of an inner component, which plays the same role as the dynamic RL module in our architecture, a query network that, given the current state, forms a query used to weight the knowledge pieces, and another fully connected layer to learn the keys of each knowledge piece. We have directly adopted the official implementation.

**KoGuN**, implemented with heterogeneous knowledge, evaluates each module and concatenates the outputs to form $k(s_t) = (\pi_{M_1}(s_t), ..., \pi_{M_K}(s_t))$. The knowledge vector $k(s_t)$ is then fed into the actor network as part of the state representation, such that $\pi(a|s_t) = \pi_\theta(k(s_t), s_t)$.

**PPO** and **SAC** serve as our standard RL baselines that do not incorporate additional knowledge.

### 6.1.3 Computational Cost

The computational cost for training and deploying AMRL is comparable with that of KoGuN. KIAN has a larger number of trainable parameters and a slightly slower inference time. PPO is faster at inference time since it does not rely on any of the knowledge modules. The details of this analysis can be found in Appendix D.

### 6.2 Heterogeneous Knowledge for Efficiency Across Environments

To evaluate how heterogeneous knowledge improves learning efficiency across a range of environments all agents are equipped with the following knowledge. In particular, we consider *Skill*, which solves the Minigrid Unlock environment; *Retrieval*, which is based on trajectory data collected in Empty Random 5x5 by an agent trained on the Empty Random 5x5 task; and, finally, *Rules*, which relies on all 6 rules for Minigrid. AMRL and baselines benefit from heterogeneous knowledge. In particular, AMRL with soft selection is more efficient and achieves higher performance than baselines across most environments (see Table 2). Not only does AMRL achieve strong final performance, but Figure 3 also shows its improved sample efficiency across several Minigrid environments. Notably, the performance difference is largest on the DoorKey 8x8 environment for which other methods have not managed to make meaningful progress by the time AMRL has solved the environment.

While AMRL with soft selection performs well, the hard selection mechanism leads to "noisy" behavior and poorer performance, often with slower convergence and without reaching competitive final performance. This should be expected due to its inability of combining the action preferences from several modules, but it is also dependent on the temperature parameter $\tau$ (see Appendix F.3). We note that all three modules are highly informative for solving the DoorKey environments in contrast to only 2 modules for Empty (Retrieval & Rules) and only 1 module for Lava Crossing (Rules). This is apparent observing the relative performance of agents accessing knowledge against PPO, which improves as the module informativeness decreases.

---

[3]The code of the libraries can be found at the following URLs: `https://github.com/lcswillems/rl-starter-files` and `https://github.com/lcswillems/torch-ac`.

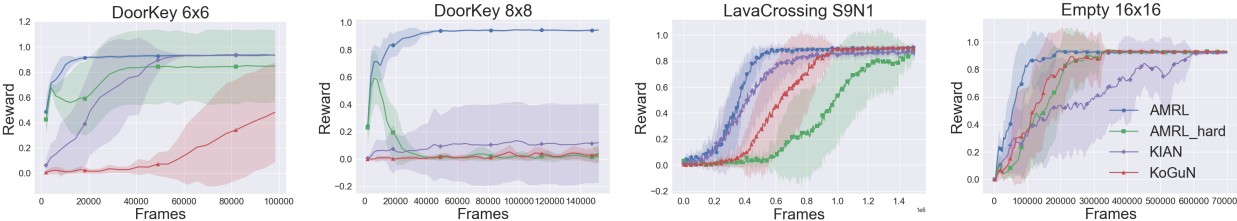

Figure 3: The achieved reward logged throughout training. On all 4 environments AMRL with soft selection uses heterogeneous knowledge to achieve good performance more efficiently than baselines. The hard selection mechanism strongly limits its capabilities and results in significantly nosier behaviors.

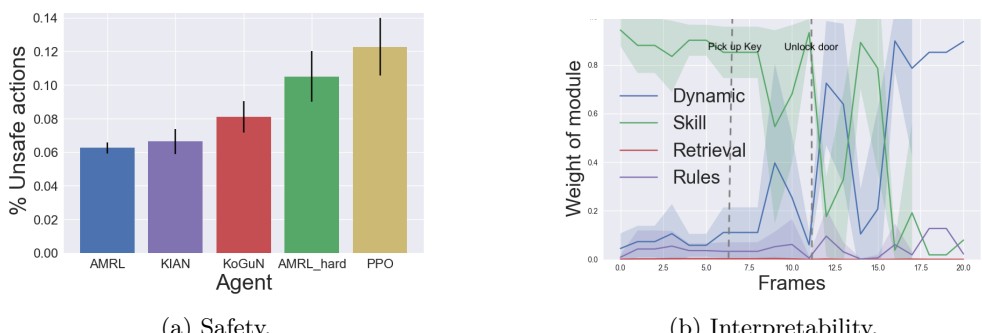

(a) Safety.                                                        (b) Interpretability.

Figure 4: 4a) Percentage of unsafe actions during training on LavaCrossing S9N1. Confidence intervals are $\pm$ 2 standard deviations across 10 random seeds. AMRL performs the smallest amount of unsafe actions closely followed by KIAN. 4b) AMRL selector weights during one episode evaluation on DoorKey 8x8 showing mean $\pm 2$ standard deviation calculated across 10 training seeds. On average, agents pick up the key with their 6th action and unlock the door with their 11th action, indicated by the dashed lines. The second milestone causes a shift in the module weights away from the unlock skill and towards the dynamic module.

## 6.3   Safer Training and Deployment

A strong motivation for utilizing heterogeneous knowledge in RL is to improve safety during training and deployment. All agents incorporating heterogeneous knowledge achieve significantly higher performance than PPO on the safety critical Lava Crossing environments, which directly corresponds to fewer unsafe actions. We record unsafe actions, such as crashing into a wall or coming in contact with the lava, during training on the LavaCrossing S9N1 environment; we plot the percentage of unsafe actions performed by each agent in Figure 4a. AMRL performs the smallest amount of unsafe actions averaged across training runs.

Beyond the above safety improvements achieved by incorporating heterogeneous knowledge, module prioritization can be deployed with AMRL for further gains. Given that a module is known to produce safe and trustworthy actions only, such as the rule module, module prioritization can be deployed to shortcut the decision making process and only act according to the safe module in critical states. This will reduce unsafe actions to 0% if the module is perfectly accurate.

## 6.4   Improving Interpretability Through Modularity

The modularity of AMRL combined with heterogeneous knowledge significantly improves interpretability. Heterogeneous knowledge sources, such as rules and skills, are inherently more interpretable than a neural network-based policy trained on the fly. Assuming interpretable modules, an analysis of the selector's outputs is sufficient to gain an insight into performed actions. Additionally, we note that the number of modules is likely to be smaller than that of the primitive actions, reducing the complexity of an analysis. Plotting the selector's weights during an evaluation episode in DoorKey (Figure 4b) shows the expected order of first relying on the Unlock skill and then the dynamic module. Additional results for other environments and agents can be found in Appendix F.2.

Table 2: Evaluation of heterogeneous knowledge for generalization and efficiency across environments. Agents use the following heterogeneous knowledge: *skill="Unlock"*, *retrieval="Empty_random_5"*, *rules="all"*. AMRL with soft selection outperforms except for the Lava S9N2 environment (note large standard deviations).

|  | DoorKey 6x6 | DoorKey 8x8 | Lava S9N1 | Lava S9N2 | Empty 16x16 |
|---|---|---|---|---|---|
| AMRL | **0.93** (0.005) | **0.94** (0.005) | **0.90** (0.009) | 0.54 (0.412) | **0.93** (0.009) |
| AMRL$_{hard}$ | 0.84 (0.285) | 0.02 (0.027) | 0.86 (0.066) | 0.02 (0.036) | **0.93** (0.010) |
| KIAN | **0.93** (0.005) | 0.11 (0.293) | 0.87 (0.021) | **0.62** (0.237) | 0.92 (0.007) |
| KoGuN | 0.32 (0.359) | 0.03 (0.040) | **0.90** (0.011) | 0.52 (0.441) | **0.93** (0.009) |
| PPO | 0.13 (0.200) | 0.01 (0.018) | 0.62 (0.357) | 0.07 (0.156) | **0.93** (0.011) |

Table 3: Mean reward (one standard deviation) with different knowledge of varying informativeness levels. Agents on DoorKey 8x8 are trained for 300k frames and on Empty 16x16 for 200k frames. The details of knowledge modules are given in Appendix E. AMRL consistently benefits from more informative modules, particularly achieving near-optimal performance in both environments. In contrast, AMRL$_{hard}$, KIAN, and KoGuN exhibit limited improvements with increasing access to informative knowledge, suggesting that they struggle to fully leverage available knowledge.

|  | DoorKey 8x8 | | | Empty 16x16 | | |
|---|---|---|---|---|---|---|
|  | low | medium | high | low | medium | high |
| AMRL | 0.03 (0.03) | **0.12** (0.28) | **0.93** (0.01) | **0.86** (0.23) | **0.84** (0.22) | **0.88** (0.18) |
| AMRL$_{hard}$ | 0.02 (0.02) | 0.03 (0.03) | 0.07 (0.06) | 0.65 (0.3) | 0.71 (0.31) | 0.65 (0.31) |
| KIAN | 0.03 (0.04) | 0.04 (0.06) | 0.27 (0.39) | 0.61 (0.35) | 0.58 (0.38) | 0.63 (0.34) |
| KoGuN | 0.02 (0.03) | 0.03 (0.028) | 0.16 (0.31) | 0.73 (0.3) | 0.79 (0.3) | 0.79 (0.27) |

## 6.5 Module Informativeness

We investigate the effect of module informativeness, i.e., how informative it is for the task at hand as discussed in Subsection 4.1, further by training agents accessing modules with high, medium, and low informativeness (details in Appendix E). The results are reported in Table 3. We find that while no agent with low module informativeness is able to solve the DoorKey 8x8 environment after 300k frames, AMRL with soft selection shows a slight advantage for medium informativeness and again significantly outperforms for high informativeness. On the Empty 16x16 environment low, medium and high knowledge has a positive impact on the agent's efficiency, but again AMRL benefits most. The larger standard deviations are due to the fact that we have stopped training earlier (200k frames) than in Table 2. The training plots for these results can be found in Appendix F.

## 6.6 Robustness against Noisy Modules

Having verified the expected effects of differences in module informativeness for a fixed number of modules, we now test the robustness of the selector against noisy modules. Specifically, we vary the density of the informative knowledge contained in all modules by adding in one, three, and five random modules, i.e., modules outputting equal action preferences for all actions. In Table 4 we report the achieved performance in terms of reward and Figure 5 shows the training runs on the LavaCrossing S9N1 environment. The training runs for the other environments can be found in Appendix F.4.

AMRL generally performs consistently well across environments, showing minimal degradation even as the number of random modules increases. AMRL$_{hard}$ shows poor performance under random modules in most environments. KIAN displays moderate robustness. Its performance degrades consistently as randomness increases. There is significant variability in performance as indicated by the larger standard deviations in some environments (e.g., Door Key 8x8). KoGuN exhibits robustness similar to AMRL, showing consistently strong performance across most environments. Additionally, Figure 5 shows that in contrast to AMRL which performs consistently, KoGuN performs well under moderate randomness, comparable to AMRL, but its sensitivity to higher levels of randomness (5 modules) reveals that it is more affected by high levels of randomness.

Table 4: Evaluating robustness in presence of random modules. Agents on LavaCrossing S9N1 for 1.5 million frames, on Empty 16x16 for 150k frames, DoorKey 8x8 for 300k frames, DoorKey 6x6 for 200k frames. We report final reward $\pm 2$ standard deviations across 10 random training seeds. AMRL is the most robust, closely followed by KoGuN. KIAN shows moderate robustness, and AMRL with hard selection is the most sensitive to random modules.

| | Lava S9N1 | | | | Empty 16x16 | | | |
|---|---|---|---|---|---|---|---|---|
| #random | 0 | 1 | 3 | 5 | 0 | 1 | 3 | 5 |
| AMRL | 0.90 | 0.89 (0.014) | 0.90 (0.011) | 0.90 (0.012) | 0.92 | 0.78 (0.264) | 0.73 (0.290) | 0.90 (0.077) |
| AMRL$_{hard}$ | 0.86 | 0.86 (0.067) | 0.21 (0.392) | 0.01 (0.013) | 0.60 | 0.22 (0.267) | 0.14 (0.108) | 0.06 (0.086) |
| KIAN | 0.87 | 0.85 (0.026) | 0.79 (0.064) | 0.75 (0.057) | 0.41 | 0.31 (0.234) | 0.25 (0.077) | 0.25 (0.104) |
| KoGuN | 0.90 | 0.90 (0.009) | 0.90 (0.016) | 0.89 (0.025) | 0.50 | 0.63 (0.341) | 0.46 (0.303) | 0.45 (0.281) |
| | DoorKey 8x8 | | | | DoorKey 6x6 | | | |
| #random | 0 | 1 | 3 | 5 | 0 | 1 | 3 | 5 |
| AMRL | 0.94 | 0.95 (0.003) | 0.57 (0.510) | 0.70 (0.404) | 0.94 | 0.94 (0.004) | 0.94 (0.004) | 0.94 (0.006) |
| AMRL$_{hard}$ | 0.04 | 0.03 (0.038) | 0.01 (0.014) | 0.02 (0.026) | 0.94 | 0.75 (0.417) | 0.44 (0.459) | 0.02 (0.030) |
| KIAN | 0.61 | 0.05 (0.069) | 0.26 (0.444) | 0.26 (0.376) | 0.93 | 0.93 (0.004) | 0.92 (0.006) | 0.92 (0.005) |
| KoGuN | 0.22 | 0.07 (0.040) | 0.04 (0.032) | 0.02 (0.024) | 0.93 | 0.93 (0.005) | 0.93 (0.005) | 0.93 (0.006) |

Figure 5: The reward dynamics during training on MiniGrid-LavaCrossing-S9N1 (training for $1.5 \times 10^6$ frames). The set of original modules is modified by adding 1, 3, and 5 additional modules outputting uniformly random actions. AMRL demonstrates consistent robustness across varying numbers of random modules, while AMRL$_{hard}$, KIAN, and KoGuN exhibit higher sensitivity, with performance and sample efficiency degrading significantly as the number of random modules increases.

Overall, AMRL is the most robust and resilient method when training in stochastic environments with random modules. It learns consistently and reaches near-optimal rewards regardless of the number of random modules. KoGuN has slightly worse performance but exhibits more variability, especially when the number of random modules is high. KIAN is moderately robust but less reliable, and AMRL$_{hard}$ is confirmed to be sensitive to randomness, almost entirely failing when random modules are introduced. This sensitivity of AMRL$_{hard}$ to random modules and its failure to scale is a direct consequence of the "all-or-nothing" nature of hard selection. When the number of modules increases, especially with the introduction of uninformative, random ones, the selector's task becomes disproportionately more difficult. It is forced to make a single correct choice from a much larger and noisier set of options with an increasingly sparse signal.

Beyond the demonstration of robustness, the results presented in Table 4 and Figure 5 can be used for assessing the scaling properties of the architecture as the number of modules increases from four modules to nine. AMRL scales generally well, while other methods struggle to identify and utilize the relevant modules.

## 6.7 AMRL in Continuous State/Action Spaces

The evaluation on the OpenAI-Robotics-Gym shown in Figure 6 shows the success rate on the FetchPickAndPlace-v1 environment with continuous state and action spaces. AMRL achieves higher performance than both SAC and KIAN, reaching near-optimal success rates within the first 200k training steps and maintaining stability throughout. In contrast, SAC plateaus around 0.7 and KIAN around 0.5, indicating limited policy improvement. The results demonstrate that when paired with SAC, AMRL significantly enhances learning efficiency and convergence, confirming its effectiveness in complex continuous control tasks.

## 6.8 Limitations and Potential Extensions

AMRL suffers from the following limitations. Firstly, it is worth noting that the complexity of the selector's action space grows with the number of modules; however, in practical settings, we expect that the number of modules will be limited. Secondly, naturally, when using existing knowledge its relevance and quality is crucial and AMRL is not an exception. However, we emphasize that diverse knowledge can contribute to developing more robust solutions, particularly in scenarios involving modules trained with noisy data or based on inconsistent rules.

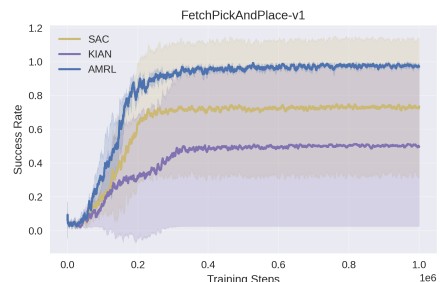

Figure 6: Success rate on the PickAnd-Place environment.

In addition, AMRL may not always offer significant advantages over simpler modular RL approaches. For example, in the limit case in which all available modules are homogeneous RL policies trained under the same paradigm, the benefits of explicit heterogeneity are absent, and the additional overhead of managing a selector outweighs the gains. Similarly, in domains where no high-quality heterogeneous knowledge is available, a standard modular RL framework or even a single-policy approach may be more practical. Future work could address these limitations by exploring adaptive mechanisms that decide when heterogeneity provides value, and by investigating scalability in scenarios with larger numbers of modules.

The release of new Large Language Models (LLM) have inspired a large body of work leveraging LLMs within RL agents, among others, to provide intrinsic rewards (Klissarov et al., 2023), guide exploration (Du et al., 2023), perform planning (Ichter et al., 2022; Huang et al., 2022), and satisfy safety constraints (Yang et al., 2021). LLMs are indeed a rich knowledge base of human preferences and as such are compatible with our proposed approach. Future work could investigate incorporating modules based on LLMs, for example, to provide high level planning abilities and additional interpretability of the system. In fact, AMRL is designed as an open, flexible architecture, capable of supporting future learning mechanisms.

# 7 Conclusion

In this paper, we have introduced Augmented Modular Reinforcement Learning (AMRL), a framework that extends modular RL to explicitly support the integration of diverse heterogeneous knowledge, such as rules, trajectory datasets, and skills. AMRL handles heterogeneity at multiple levels: knowledge sources (rules, skills, retrieval, dynamic RL), state representations, and processing mechanisms. Through a unified module abstraction and a representation-agnostic selector, it allows diverse modules to be composed seamlessly.

We have formally characterized heterogeneous knowledge and investigated several sources of information heterogeneity. We have implemented AMRL in environments from the Minigrid and the OpenAI-Robotics-Gym suites, benchmarking against KIAN, KoGuN, PPO and SAC. Our results show that AMRL consistently improves sample efficiency, achieves safer exploration, and enhances robustness in the presence of noisy modules. Moreover, by structuring decision-making around modular and interpretable components, AMRL increases the transparency of agent behavior and facilitates generalization across tasks. We also analyzed hard versus soft selection, demonstrating that soft selection is particularly effective when integrating heterogeneous modules.

**Broader Impact Statement**

AMRL is motivated by the goal of improving safety and robustness in reinforcement learning by enabling agents to incorporate heterogeneous knowledge such as rules, skills, and retrieval mechanisms. This allows for safer exploration, better interpretability, and more reliable behavior in domains where purely trial-and-error learning can be unsafe. However, the safety and reliability of AMRL ultimately depend on the quality of the incorporated knowledge. Flawed, outdated, or biased rules and demonstrations could propagate harmful behaviors, and users may overestimate the guarantees provided by modular integration. While modularity enhances transparency relative to end-to-end RL, responsible deployment requires careful validation of knowledge sources and explicit consideration of ethical risks associated to it.

**Acknowledgments**

Lorenz Wolf was supported by the UK Engineering and Physical Sciences Research Council (EP/S021566/1). Mirco Musolesi acknowledges the support of the UK Engineering and Physical Sciences Research Council through the grant EP/X028569/1.

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

# A  Policy Gradients for AMRL

## A.1  Notation

Given a policy $\pi$ such that $\pi(a, s) = Pr(a|s)$ is the probability of taking action $a$ in state $s$ when following policy $\pi$ and assuming we have an finite set of actions $\mathcal{A} = \{a_i\}_{i=1}^{|\mathcal{A}|}$, we use $\pi(s)$ to denote the vector with ith component being $\pi(a_i|s)$. Additionally, given a list of modules $\mathbf{M}$ with corresponding policies $\{\pi_M\}_{M \in \mathbf{M}}$ we use $\mathbf{\Pi_M}(a|s)$ to denote the column vector with ith entry $\pi_M(a|s)$ where $M$ is the ith module in $\mathbf{M}$.

Furthermore, denote the selectors policy parameterized by $\phi$ as $\pi_{selector}^\phi(\cdot|s)$ and the policy of the dynamic module as $\pi_{dyn}^\theta(\cdot|s)$. Let the set $\mathbf{M}$ contain all modules of which one is the dynamic module.

## A.2  Policy Gradients

The vanilla Policy Gradient of a policy $\pi^\theta$ is given by:

$$\nabla_\theta J\left(\pi^\theta\right) = \operatorname*{E}_{\tau \sim \pi^\theta}\left[\sum_{t=0}^{T} \nabla_\theta \log \pi^\theta\left(a_t \mid s_t\right) A^{\pi^\theta}\left(s_t, a_t\right)\right], \tag{6}$$

where $J$ is the un-discounted finite time horizon reward to-go. Furthermore, the update rule used in PPO (Schulman et al., 2017) is given by:

$$\theta_{k+1} = \arg\max_\theta \frac{1}{|\mathcal{D}_k| T} \sum_{\tau \in \mathcal{D}_k} \sum_{t=0}^{T} \min\left(\frac{\pi_\theta\left(a_t \mid s_t\right)}{\pi_{\theta_k}\left(a_t \mid s_t\right)} A^{\pi_{\theta_k}}\left(s_t, a_t\right), \quad g\left(\epsilon, A^{\pi_{\theta_k}}\left(s_t, a_t\right)\right)\right). \tag{7}$$

While the below analysis is focused on the vanilla policy gradient, it justifies the use of PPO to train the selector.

## A.3  Soft Selection

Using the specified notation we can write the AMRL policy with soft selection as:

$$\pi^{\phi,\theta}(a|s) = \pi_{selector}^\phi(s)^T \mathbf{\Pi_M}(a|s) \tag{8}$$

where $\mathbf{M}$ is the set of modules, of which the only module parameterized is the dynamic module $M_{dyn}$ with policy $\pi_{dyn}^\theta$. Alternatively, we can write:

$$\pi^{\phi,\theta}(a|s) = \pi_S^\phi(M_{dyn}|s)\pi_{M_{dyn}}^\theta(a|s) + \sum_{M \in \mathbf{M} \setminus M_{dyn}} \pi_S^\phi(M|s)\pi_M(a|s), \tag{9}$$

where we have written out the inner product and split out the term corresponding to the dynamic module.

We can compute the policy gradients with respect to each of the parameter sets $\phi$ and $\theta$ as follows. The gradient w.r.t. the dynamic module parameters $\theta$ can be computed as:

$$\nabla_\theta J(\pi^{\phi,\theta}) = \operatorname*{E}_{\tau \sim \pi_\theta}\left[\sum_{t=0}^{T} \nabla_\theta \log \pi^{\phi,\theta}\left(a_t \mid s_t\right) A^{\pi^{\phi,\theta}}\left(s_t, a_t\right)\right] \tag{10}$$

$$\tag{11}$$

We can then derive:

$$\nabla_\theta \log \pi^{\phi,\theta}\left(a_t \mid s_t\right) = \frac{1}{\pi^{\phi,\theta}\left(a_t \mid s_t\right)} \nabla_\theta \pi^{\phi,\theta}\left(a_t \mid s_t\right) \tag{12}$$

$$= \frac{1}{\pi^{\phi,\theta}\left(a_t \mid s_t\right)} \pi_S^\phi(M_{dyn}|s_t) \nabla_\theta \pi_{dyn}^\theta(a_t|s_t). \tag{13}$$

Similarly, we compute the policy gradient w.r.t. the selector's parameters $\phi$, as follows:

$$\nabla_\phi \log \pi^{\phi,\theta}\left(a_t \mid s_t\right) = \frac{1}{\pi^{\phi,\theta}\left(a_t \mid s_t\right)} \nabla_\phi \pi^{\phi,\theta}\left(a_t \mid s_t\right) \tag{14}$$

$$= \frac{1}{\pi^{\phi,\theta}\left(a_t \mid s_t\right)} \nabla_\phi \boldsymbol{\pi_{selector}^\phi}(s)^T \boldsymbol{\Pi_M}(a|s) \tag{15}$$

$$= \frac{1}{\pi^{\phi,\theta}\left(a_t \mid s_t\right)} \left( \nabla_\phi \pi_S^\phi(M_{dyn}|s_t)\pi_{dyn}^\theta(a_t|s_t) + \sum_{M \in \mathbf{M}} \nabla_\phi \pi_S^\phi(M|s_t)\pi_M(a_t|s_t) \right). \tag{16}$$

$$\tag{17}$$

### A.4 Hard Selection

In the case of hard selection we use the Gumbel Softmax trick to perform differentiable selection. In particular, we want to sample from a categorical distribution where the probabilities are given by the selector as $\boldsymbol{\pi_S^\phi}(s)$. In other words, we select module $M$ with probability equal to $\pi_{selector}^\phi(M|s)$. We then obtain a sample vector $\mathbf{y}$ of dimensions $|\mathcal{A}_{selector}|$ by setting:

$$\mathbf{y}_i = \frac{\exp\left(\left(\log\left(\pi_{selector}^\phi(M_i|s)\right) + g_i\right)/\tau\right)}{\sum_{j=1}^k \exp\left(\left(\log\left(\pi_{selector}^\phi(M_j|s)\right) + g_j\right)/\tau\right)} \quad \text{for } i = 1, \ldots, k, \tag{18}$$

where $g_1 \ldots g_k$ are i.i.d samples drawn from Gumbel$(0,1)$, which can be easily sampled using an inverse transform. Note that the softmax function is used as a continuous differentiable approximation. As the temperature $\tau$ tends to 0, the samples $\mathbf{y}$ tend to the samples of the true categorical distribution Jang et al. (2017).

## B Environments

In all the below environments the action space consists of 7 discrete actions shown in Table 5. The observation size is 7x7 and is marked by the light grey grid centered around the agent in Figure 7.

Table 5: Action space of the Minigrid environments.

| Num | Name | Action |
|-----|------|--------|
| 0 | left | Turn left |
| 1 | right | Turn right |
| 2 | forward | Move forward |
| 3 | pickup | Unused |
| 4 | drop | Unused |
| 5 | toggle | Unused |
| 6 | done | Unused |

## C Additional Implementation Details

For all agents below the PPO hyperparameters are set to the default values provided in the `rl-starter-file` repository (https://github.com/lcswillems/rl-starter-files).

**PPO.** We use the network architecture of actor and critic network as implemented in the `rl-starter-files` repository. In particular:

- *Embedding*: A CNN with 3 convolutional layers produces an embedding of the image observations. Both the actor and the critic rely on these image embeddings.

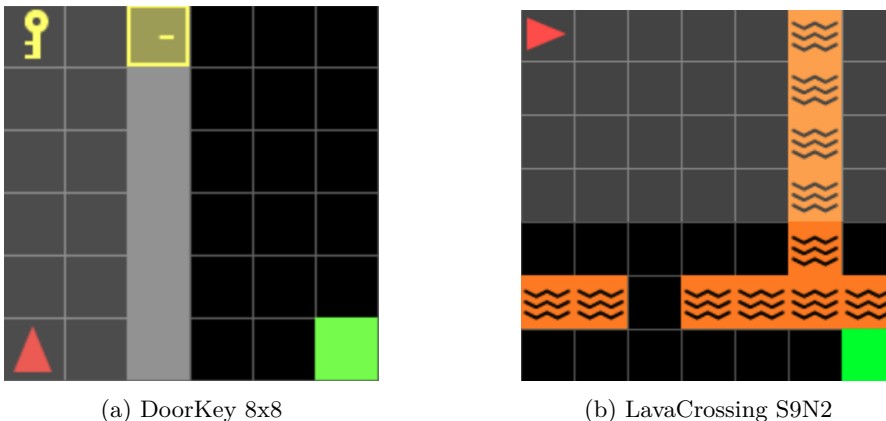

(a) DoorKey 8x8        (b) LavaCrossing S9N2

Figure 7: Snapshots taken from two of the Minigrid environments.

- *Actor*: The actor network is a 2 layer fully connected neural network with layer width 64.

- *Critic*: The critic network is a 2 layer fully connected neural network with layer width 64.

**AMRL.** In the following, we summarize the key design and implementation choices of our algorithm:

- *Embedding*: Same as in PPO. The Embedding network is shared by the selector, the dynamic RL module, and the critic network.

- *Selector*: The selector network is a 2 layer fully connected neural network with layer width 64. It outputs a weight for each module. The softmax function is applied to the weights to map them to probabilities. To sample a module in the hard selection mechanism we use the Gumbel softmax distribution.

- *Critic*: Same as in PPO.

**KIAN.** We rely on the original implementation provided by the authors: `https://github.com/Pascalson/KGRL`

- *Embedding*: A CNN with 3 convolutional layers produces an embedding of the image observations. The inner actor, the key network, and the critic rely on these image embeddings.

- *Actor*: The actor consists of an internal actor, the query network, and a key network. All 3 share an initial fully connected base layer of width 64, which is followed by an additional fully connected layer mapping from 64 features to the desired dimension (embedding size of 8 for the query and key network, and $|\mathcal{A}|$ for the internal actor). The keys of modules are learned by the nn.Embedding layer.

- *Critic*: Same as in PPO.

**KoGuN,** The following summarizes the key design choices of the KoGuN implementation.

- *Embedding*: Same as for PPO. Both the actor, and the critic rely on these image embeddings.

- *Actor*: The actor network is a 2 layer fully connected neural network with layer width 64. It takes as inputs the output of the embedding network concatenated with the action preferences averaged across modules.

- *Critic*: Same as in PPO.

**Modules.** All modules are implemented as described in the body of the main paper.

**Compute Resources.** The experiments were run on a CPU. No large amount of memory is required.

## D Computational Cost

An analysis of the computational cost of AMRL and baselines can be found in Table 6.

Table 6: Analysis of the computational cost. The number of trainable parameters includes the trainable dynamic module for AMRL and the inner actor for KIAN. All knowledge based architectures are comparable and rely on the skill, retrieval and rule modules. The retrieval module searches for 4 neighbors in a dataset of 3040 transitions. The faster inference times of PPO are caused by not calling the knowledge modules.

| Method | # Trainable params | Total inference time (s) |
|--------|--------------------|--------------------------|
| PPO    | 19384              | $3.4 \times 10^{-5}$     |
| AMRL   | 19644              | $1.5 \times 10^{-4}$     |
| KoGuN  | 19832              | $1.5 \times 10^{-4}$     |
| KIAN   | 20448              | $1.6 \times 10^{-4}$     |

## E Additional Experimental Details

**DoorKey 8x8.** The knowledge configurations of different levels used for the DoorKey 8x8 environment is as follows:

- *Low:* Skill - Empty 5x5, Retrieval - Empty 5x5, Rules - No rules.

- *Medium*: Skill - DoorKey 5x5, Retrieval - DoorKey 5x5, Rules - All rules.

- *High:* Skill - DoorKey 6x6, Retrieval - DoorKey 6x6, Rules - All rules.

**Empty 16×16.** On Empty, the knowledge for different informativeness levels is:

- *Low:* Skill - Empty 5x5, Retrieval - Empty 5x5, Rules - No rules.

- *Medium:* Skill - Empty 8x8, Retrieval - Empty 8x8, Rules - All rules.

- *High:* Skill - Empty 16x16, Retrieval - Empty 16x16, Rules - All rules.

## F Additional Results

### F.1 Reward Plots

For completeness, we present the training runs for the module informativeness experiment in Figures 8 and 9.

### F.2 Analyzing Learned Policies

To further investigate the learned policies and showcase improved interpretability, we plot the weights of the Selector in AMRL and the weights of the attention mechanism in KIAN to compare the learned policies in a range of selected environments. To obtain the weights, the agents are evaluated in the specified environments, weights are recorded for one episode (same configuration) from start to finish for all training seeds and we report mean weights together with $\pm 2$ standard deviation confidence intervals. Note that weights are not comparable across different start positions and environment instances which is why we reported weights for the same episode. The extracted weights are shown in Figure 10.

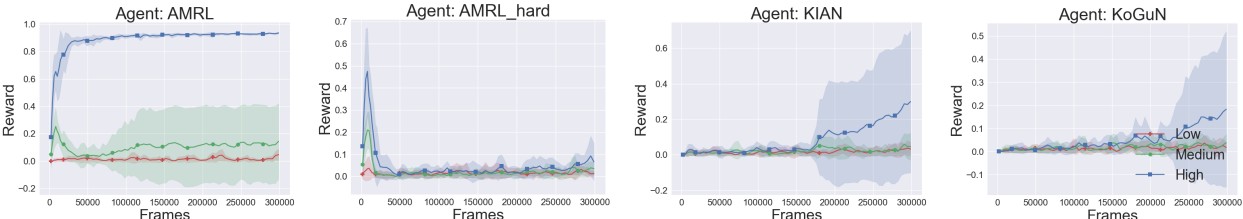

Figure 8: Average training run performance on DoorKey 8x8 with different levels of knowledge.

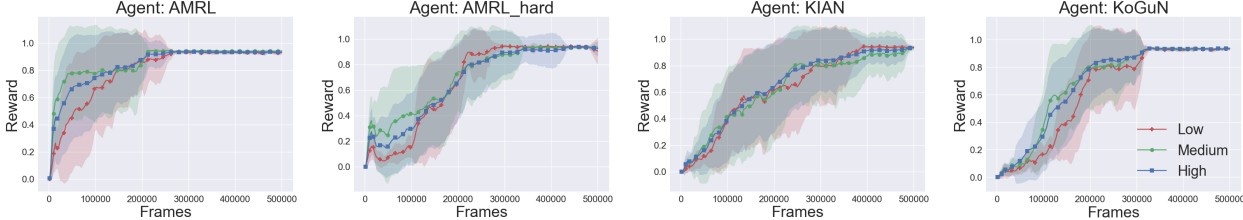

Figure 9: Average training run performance on Empty 16x16 with different of levels of knowledge. After 500k all agents have converged to roughly equal performance. We shorten the training runs and instead reported the performance after 200k frames of training.

### F.3 Gumbel Temperature Ablation

We perform an ablation on the temperature parameter of the Gumbel-Softmax distribution used for the hard selection mechanism. The results of the ablation study are reported for the following two cases: 1) the samples are discretized and the smooth approximation is used for gradients as in the experiments reported in the paper; 2) the samples are not discretized.

For both cases we train for 100k frames on the DoorKey $6{\times}6$ environment and plot the mean rewards with confidence intervals across 10 random training seeds. The results are shown in Figures 11.(a) and 11.(b) for the discretized and not discretized cases respectively. We find that larger $\tau$ around 3 performs best in both cases. Additionally, the performance drops as $\tau$ decreases to 0 and the samples from the categorical distribution become more and more discrete regardless of whether we discretize samples to a one-hot vector in the forward pass.

### F.4 Presence of Random Modules

In Figures 12, 13, and 14 we report the training curves on the DoorKey 8x8, DoorKey 6x6, and Empty 16x16 environments, for evaluating robustness in presence of random modules.

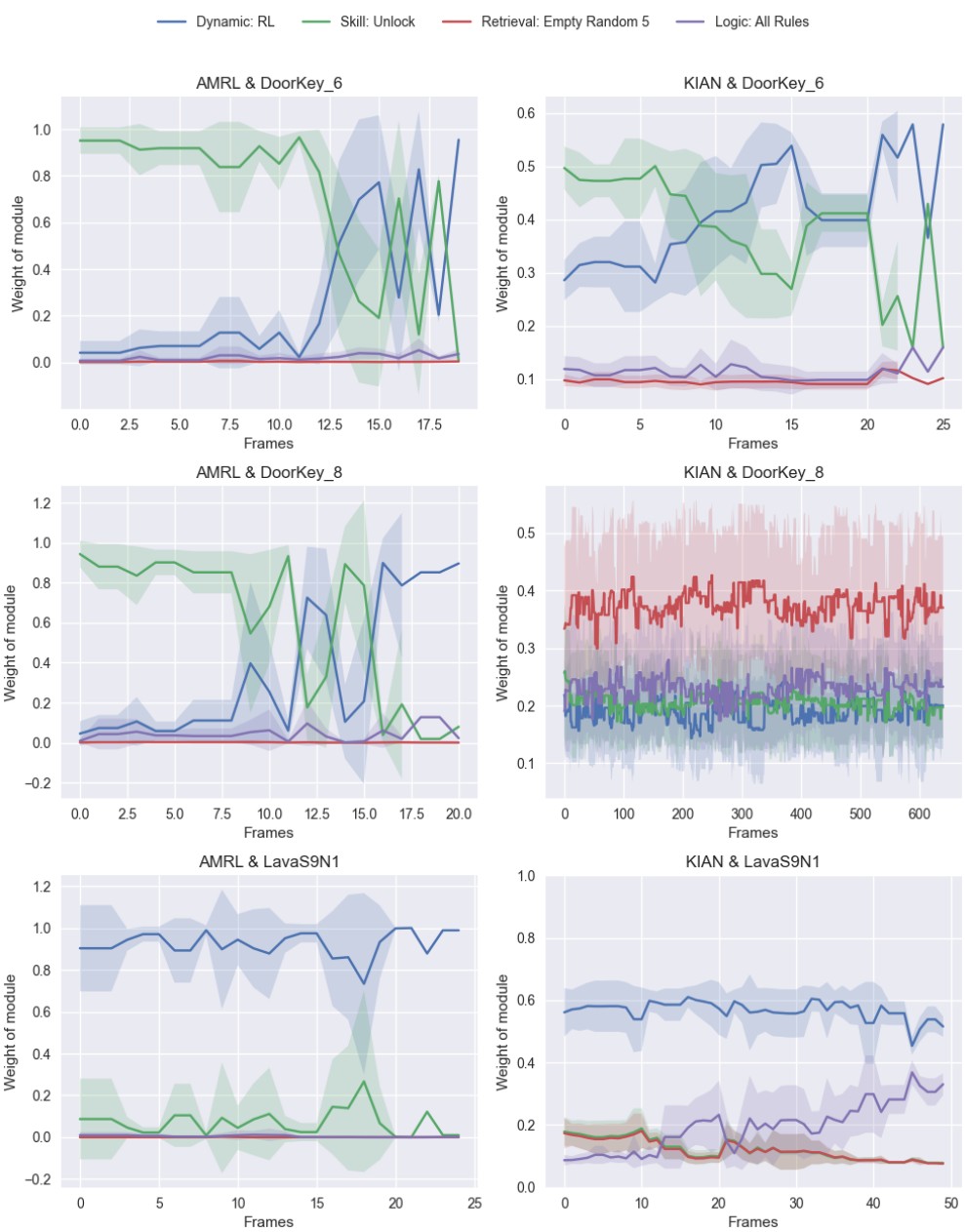

Figure 10: Weightings of modules in the final policy. Note that in contrast to AMRL, KIAN has not converged on the DoorKey 8 × 8 environment and seems to be stuck because it puts too much weight on the retrieval module instead of the more relevant skill module. In the LavaCrossing S9N1 environment, AMRL has converged to almost fully rely on the dynamic RL module in contrast to KIAN, which has slightly more balanced weights.

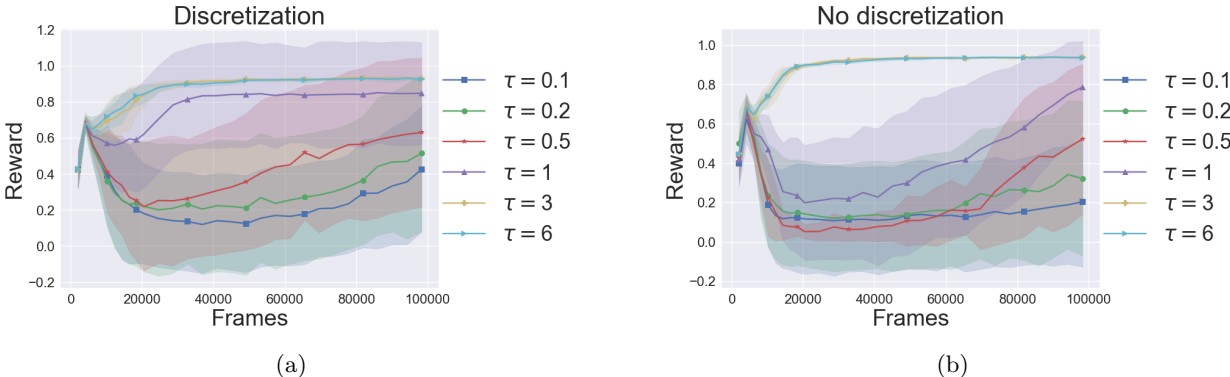

Figure 11: (a) Ablation on the temperature with discretization in the forward pass. (b) Ablation on the temperature without discretization in the forward pass.

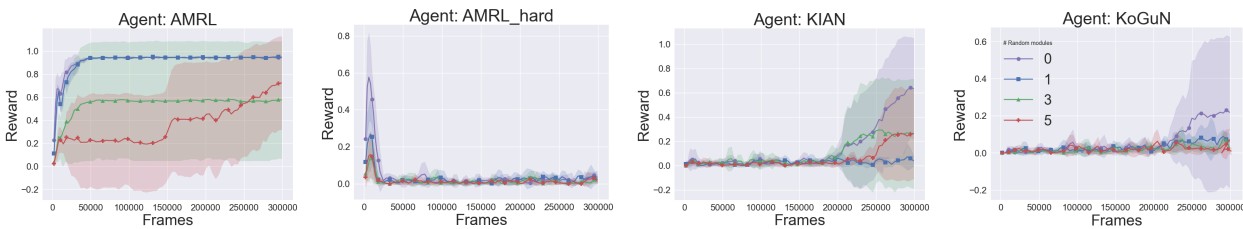

Figure 12: The achieved reward logged throughout training on Door Key 8x8. The set of original modules is modified by adding 1, 3, and 5 modules outputting uniformly random actions. It shows that AMRL can solve the environment with random modules present. However, training is slower for 3 and 5 random modules added. The results for other agents show significantly worse performance when random modules are present, but results are less conclusive due to significantly worse learning in the first place.

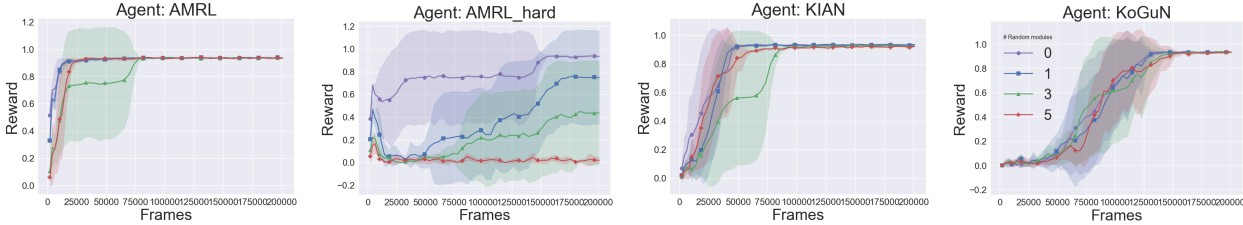

Figure 13: The achieved reward logged throughout training on Door Key 6x6. The set of original modules is modified by adding 1, 3, and 5 modules outputting uniformly random actions.

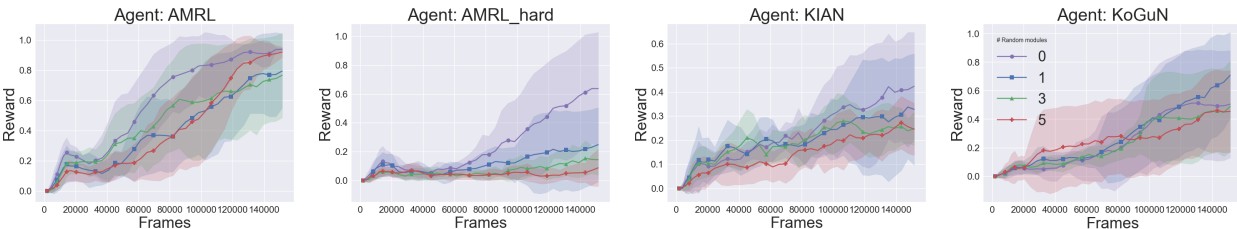

Figure 14: The achieved reward logged throughout training on MiniGrid-Empty-16x16, training for 150k frames. The set of original modules is modified by adding 1, 3, and 5 modules outputting uniformly random actions.

