# OpenReview forum: "Heterogeneous Knowledge for Augmented Modular Reinforcement Learning"
_TMLR — Accepted by TMLR_

### Review · Reviewer_3asA · 2025-06-23

**Summary Of Contributions:**

This paper presents a modular / hierarchical reinforcement learning algorithm that is able to combine heterogeneous sources of information, such as rules, expert trajectories, and skills trained previously (on possibly unknown tasks). The main idea is to train a policy to select a module (or a mixture thereof) and then take the actions recommended by that module. The method is illustrated on different tasks in the GridWorld environment, showing improved reward learning compared to previous methods, increased safety, and interpretability.

**Audience:**

Yes

**Broader Impact Concerns:**

No broader impact statement is provided. However, since the paper states that the increased safety due to the usage of rules is one of the main advantages of the algorithm, I feel that one should be included.

**Claims And Evidence:**

Yes

**Requested Changes:**

### Critical ###

- Add experiments on a different environment, if possible.

**Strengths And Weaknesses:**

### Strengths ###

- The paper is well-written and straightforward to understand.
- The proposed algorithm is simple and technically sound.
- Experiments show that compared to baselines, the proposed method leads to higher reward and increased safety. It is also more robust to noisy sources of knowledge.

### Weaknesses ###

- Experiments were done on a fairly simple environment. GridWorld has a limited state space and action space.
- It is not clear how rules could be incorporated in a more complex environment. For example, for a legged robot a rule could be to jump when there is an obstacle in front; how can this be translated into a policy recommendation?
- No code is provided.

---

> ### Author Response · Authors · 2025-07-09
> **Clarification regarding extension to an additional environment**
>
> Dear Reviewer,
>
> We are very grateful for your time and for the insightful and constructive feedback on our manuscript.
> We are preparing a comprehensive response and we will post it once we will receive all the reviews.
>
> In the meanwhile, we are working on additional experiments as requested. To ensure we proceed in the most effective direction, we would appreciate a point of clarification regarding potential experimental environments.
>
> We are working on extending our experiments and are considering MiniHack (https://minihack.readthedocs.io/en/latest/) as additional environment, as suggested. We would be grateful for your perspective on it. Especially, we were wondering if you might consider it as sufficiently insightful as additional environment for our performance evaluation or if you might want to suggest a more suitable environment.
>
> We value your guidance as we work to improve our submission.
>
> Respectfully,
>
> The Authors

---

> > ### Author Response · Authors · 2025-08-04
> > **Follow-up**
> >
> > Dear Reviewer,
> >
> > In order to address the concerns raised in the review, we would like to bring our above question to your attention.
> >
> > Thanks again for reviewing our work; we fully understand the competing demands of reviewers in this period of the year.
> >
> > Sincerely,
> >
> > The Authors

---

> > > ### Comment · Reviewer_3asA · 2025-08-31
> > >
> > > Hello authors,
> > >
> > > I apologize for the delay in replying. After taking a look at MiniHack, I think it looks promising, in particular the skill acquisition tasks, which have a substantially larger action space.
> > >
> > > Best,
> > > Reviewer 3asA

---

### Review · Reviewer_gHgt · 2025-08-17

**Summary Of Contributions:**

The paper introduces Augmented Modular Reinforcement Learning (AMRL), a framework that extends traditional modular RL by integrating heterogeneous knowledge modules. Unlike standard approaches where sub-policies are typically homogeneous RL policies, AMRL allows modules to be derived from diverse sources, including rule-based logic, trajectory retrieval, pre-trained RL skills, and dynamic RL policies. A selector mechanism is proposed to arbitrate among these modules, either through hard selection (Gumbel-softmax sampling) or soft selection (weighted fusion).

The contributions claimed are:

1. Formalization of heterogeneous modules for decision-making, providing a unified representation that accommodates different types of knowledge sources and processing mechanisms.
2. Design of AMRL, an augmented modular RL framework with a selector that can flexibly combine heterogeneous modules in a plug-and-play manner.
3. Empirical evaluation on MiniGrid environments, showing that AMRL with soft selection improves sample efficiency, safety, and robustness compared to baselines (PPO, KoGuN, KIAN).
4. Discussion of how heterogeneity enhances interpretability and safety, and how the framework can potentially incorporate future knowledge sources such as LLMs.

**Audience:**

Yes

**Claims And Evidence:**

Yes

**Requested Changes:**

### Critical

1. Clarify the causal motivation.
   The current manuscript does not provide a precise explanation of why prior HRL/modular RL methods cannot already integrate heterogeneous sub-policies. The paper should articulate clearly:

   * What practical or theoretical barriers exist in prior frameworks (e.g., comparability of reward scales, incompatibility of knowledge representations)?
   * Why these barriers motivate AMRL specifically, rather than simply applying existing arbitration/fusion techniques.
     Strengthening this causal chain is crucial to making the contribution well-justified.

2. Clarify the role of state representations.
   All modules in AMRL appear to consume the same environment state (e.g., 7×7 grid observation) and produce distributions over the same action space. If this is indeed the case, the heterogeneity is only in internal processing, not in state or action representation. The authors should explicitly discuss:

   * How each module interprets the state, and whether any transformations or abstractions are applied.
   * Why this setup should be regarded as genuinely heterogeneous rather than a collection of differently implemented policies.

### Important but not Critical

3. Position the novelty more carefully.
   While the paper emphasizes heterogeneity as a new angle, prior HRL frameworks did not forbid heterogeneous modules; they simply did not emphasize them. The paper would be stronger if it acknowledged this, and reframed the novelty as:

   * Providing a unified formalization of heterogeneous modules;
   * Demonstrating empirically that such integration improves sample efficiency, safety, and robustness.
     This more modest but accurate positioning would make the contribution clearer and more defensible.

4. Refine discussion of hard vs. soft selection.
   The current exposition could better situate these mechanisms in relation to existing work (e.g., command arbitration/fusion, mixture-of-experts). Rather than presenting them as new, the paper should highlight how they behave differently in the presence of heterogeneous modules, and why soft selection may be preferable.

### Optional (would strengthen the paper)

5. Expand discussion of limitations.
   The paper already notes scaling issues with module count and reliance on knowledge quality. It would further help readers if the authors provided concrete scenarios where AMRL may not offer advantages over simpler modular RL (e.g., when all modules are homogeneous policies).

6. Tighten writing for causal clarity.
   Some passages emphasize breadth and polish over clarity (likely with automated editing). The introduction and related work would benefit from explicitly linking:

   * Identified limitations of prior methods →
   * Why those limitations matter in practice →
   * How AMRL specifically addresses them.

**Strengths And Weaknesses:**

### Strengths

* Clear motivation: The paper highlights the limitations of standard modular RL frameworks, which generally assume homogeneous RL-based sub-policies, and argues for the need to handle heterogeneous knowledge sources.
* Unified framework: The proposed AMRL architecture provides a simple but flexible way to incorporate diverse module types (rules, retrieval, skills, dynamic RL) through a common selector. This framing emphasizes plug-and-play extensibility, which may inspire broader adoption.
* Selector formalization: The paper systematically contrasts hard selection (Gumbel-softmax arbitration) and soft selection (weighted fusion), situating them within prior work on command arbitration and command fusion.
* Empirical demonstration: Experiments on MiniGrid illustrate benefits in terms of sample efficiency, safety, and robustness, especially when combining informative heterogeneous modules. The use of safety metrics (e.g., unsafe actions in LavaCrossing) and interpretability analysis (selector weights) goes beyond simple reward reporting.
* Discussion of extensions: The paper anticipates future directions, including the integration of LLM-based modules, which shows awareness of broader trends in RL + knowledge integration.

### Weaknesses

* Unclear novelty relative to prior HRL/modular RL:
  The paper frames heterogeneous knowledge integration as something fundamentally new, but it is not convincingly argued why existing HRL or modular RL frameworks could not already accommodate sub-policies with different internal implementations. Traditional frameworks did not forbid heterogeneity; they simply assumed homogeneous policies for convenience. Thus, the conceptual novelty appears incremental.

* Vague causal link between problem and solution:
  The manuscript does not provide a precise explanation of what prevents prior methods from directly combining heterogeneous modules. The claimed “gap” is described mostly at a high level, without identifying specific theoretical or representational incompatibilities. The proposed solution — treating each module as producing an action preference and aggregating them via a selector — is reasonable but does not clearly resolve a previously insurmountable difficulty.

* Ambiguity in state representation:
  All modules ultimately consume the same state observation (MiniGrid grid view) and output distributions over the same action space. The heterogeneity lies only in their internal processing (rules, retrieval, RL policy), not in their input or output interfaces. This raises the question of whether AMRL introduces genuine heterogeneity or simply rephrases existing modular RL as a mixture of differently implemented policies. The paper does not explain how states would need to be transformed into different sub-states for modules that conceptually require distinct input formats.

* Selector design is not fundamentally new:
  The distinction between hard and soft selection largely mirrors well-studied paradigms (command arbitration vs. fusion, mixture-of-experts). The technical contribution of adopting Gumbel-softmax for differentiability and weighted averaging for fusion is modest, and their relationship to existing approaches should be acknowledged more explicitly.

* Writing and positioning issues:
  While the manuscript is well-polished (likely with LLM assistance), the narrative sometimes emphasizes breadth at the expense of causal clarity. The problem statement, contribution claims, and experimental design are not always tightly connected, leaving the reader uncertain whether AMRL solves a genuine limitation or simply demonstrates a repackaging of familiar ideas.

---

### Review · Reviewer_H5ND · 2025-08-31

**Summary Of Contributions:**

This paper propose a model to incorporate heterogeneous knowledge into the  Augmented Modular Reinforcement Learning method by establish policy $\pi_M(s)$ for $M^{th}$ knowledge source and introduce a selector network that learns to decide the weight of each knowledge source based action that linearly combines to the final action.

**Audience:**

Yes

**Claims And Evidence:**

Yes

**Requested Changes:**

1. Provide explanation for Fig. 5 AMRL failure cases.
2. If possible, test some continuous state/action space cases.

**Strengths And Weaknesses:**

Pros:
1. Method looks interesting
2. Paper easy to follow
3. Sufficient ablation study

Cons:
1. The experiments are for discrete action and thus finite state space, a experiment with continuous action/state space could provide a much stronger empirical support for the model.
2. In Fig. 5, the AMRL hard looks worse in some cases, where 5 random modules result in flat reward, this should be further analyzed.
3. I could be nitpicking, but the title looks a bit exaggerating. The proposed model expect each source of the "heterogeneous knowledge" to yield a corresponding policy. In practice, this may not be implemented easily, because human knowledge are not always "well defined".

---

### Comment · Action_Editor_QCjN · 2025-12-04
**Content is more than 12 pages**

Dear authors, you are now at 14 pages. Can you cut it down to 12 since you submitted this under the "Regular submission (no more than 12 pages of main content)" option?

Thank you

---

> ### Comment · Action_Editor_QCjN · 2025-12-04
> **14 pages is fine.**
>
> Sorry for the confusions, 14 pages is fine.

---

### Decision · Action_Editor_QCjN · 2025-10-20

**Recommendation:** Accept as is

**Audience:**

Yes

**Audience Explanation:**

Members of the community that are interested in either hierarchical RL or logic would find this interesting.

**Claims And Evidence:**

Yes

**Claims Explanation:**

Reviewers agreed that the claims are supported, especially after the revision.